# Educational Pathways, Spatial Skills, and Academic Achievement in Graphic Expression in First Year of Engineering

José Sebastián Velázquez [1], Francisco Cavas [1], María Castillo Fuentes [2] and Rafael García-Ros [3,*]

1    Department of Structures, Construction And Graphic Expression, Polytechnic University of Cartagena, 30202 Cartagena, Spain; jose.velazquez@upct.es (J.S.V.); francisco.cavas@upct.es (F.C.)
2    Department of Behavioral Sciences Methodology, University of Valencia, 46010 Valencia, Spain; m.castillo.fuentes@uv.es
3    Department of Developmental and Educational Psychology, University of Valencia, 46010 Valencia, Spain
*    Correspondence: rafael.garcia@uv.es

**Abstract:** The subject of Graphic Expression, which is mandatory in the first year of engineering studies, showed poor academic results in recent years. This study analyzes the relationship and predictive capacity of various variables that previous research highlighted as relevant: prior academic preparedness, educational itinerary followed, mental rotation skills, videogame usage, as well as the gender and age of the students. A total of 161 first-year engineering students from a technical university in southern Europe participated in the study. Their spatial rotation skills were evaluated using the MRT-A, gathering information about the rest of the relevant variables and obtaining their academic results at the end of the course. The predictive capacity of the variables on academic performance was determined through linear regression techniques (grade in the subject, on a 0–10 scale) and logistic regression (pass/fail). All variables are significantly related to academic results in the expected direction, except for videogame usage and gender. No significant differences in spatial skills were found between genders, although differences were observed in videogame usage. The best predictors of performance are prior preparation and the educational itinerary followed. The results are discussed considering previous research, highlighting measures to improve results in Graphic Expression, with emphasis on training in spatial skills.

**Keywords:** engineering studies; academic preparedness; gender differences; university students; mental rotation skills; Technical Drawing

## 1. Introduction

Engineering students face notable difficulties in achieving satisfactory academic results in their degree programs, a fact that is particularly evident in their first year of enrollment if both academic performance and dropout rates are put under scrutiny. Similarly to what happens at the international level, in the Spanish context, STEM studies (science, technology, engineering, and mathematics) are the ones that exhibit higher dropout rates, ranging from 37.4% to 49.9% of students [1], more than half of them occurring in the first year of the studies [2]. At the Technical University of Cartagena (UPCT), where this study focuses on its engineering degrees, dropout rates are even higher than the national average [3]. In the 2018–2019 cohort, a dropout rate of 24% was observed in the first year of university enrollment, while the academic performance, defined as the percentage relationship between the number of credits actually passed by students and the total enrolled credits during that academic year, of the 2021–2022 cohort reached only 50%, creating an enabling environment for early dropout. More specifically, the high rates of academic failure in the recent cohorts entering the subject Graphic Expression (around 50% of the students in Mechanical Engineering (GIM), Design Engineering (GIDI), and double degree in Mechanical and Design Engineering (PCEO) participating in this study),

a mandatory first year subject of engineering studies, on which this study focuses, led us to carry out this work.

The reasons for these poor academic results and high dropout rates in the first year of university enrollment are diverse and multifactorial. Previous research identified the relevance of sociodemographic factors (e.g., age of entry and socioeconomic level), prior education (e.g., academic performance and educational background), and socio-personal factors (e.g., cognitive and non-cognitive abilities, personality, learning strategies, stress coping and emotional regulation strategies, expectations, and motivation) of students, as well as institutional factors (e.g., university support services and teaching methodology) and factors related to students' experiences once they are enrolled in university (e.g., academic and social integration, as well as institutional commitment) [4–10]. A comprehensive longitudinal study conducted with Spanish university students [11], highlights that prior academic preparedness (pre-enrollment academic ability), age of university entry, family characteristics, and previous educational experience are the best predictors of university persistence/dropout. Other studies conducted in the Spanish context also emphasize the importance of university entrance scores and academic results in the first year, along with excessive workload and inadequate time and stress management skills of students [12–14]. Various studies with engineering students also show that both the grades obtained in the first year of the degree and institutional commitment are excellent predictors of persistence in sophomore year [2,15,16], and that university entrance scores significantly influence persistence/dropout through the former [2]. Lastly, numerous studies highlight the crucial role of spatial abilities in achieving satisfactory academic results in engineering studies [17–19], with some of them finding that those who drop out of these studies, on average, have lower spatial competence than their peers who remain in them [20].

### 1.1. Spatial Skills in Engineering Studies and in Graphical Expression

Previous research highlights that the most suitable cognitive profile for STEM students goes beyond high general reasoning ability, emphasizing that mathematical competence and spatial capacity are fundamental cognitive requirements for learning in these fields [17,19]. Moreover, longitudinal studies analyzing the development trajectories of large samples of students over two decades found that spatial skills are significant predictors of STEM achievement and attainment, with their unique contribution to students' academic outcomes remaining significant even after controlling for the effects of their mathematical and verbal abilities [21–23]. Thus, spatial skills, particularly the mental rotation skills considered in this study, are significantly related to academic results in various STEM disciplines (e.g., mathematics, physics, chemistry, and geology), especially in engineering studies [17,19,22,24]. For example, Sorby [19] developed a spatial workbook for engineering students that focused on 3D visualization skills, finding that students who used it improved their spatial visualization abilities and significantly increased their likelihood of achieving satisfactory academic results. In a subsequent study, Sorby et al. [24] assigned engineering students with lower spatial visualization skills to a training course, observing not only improvements in their spatial abilities, but also higher calculus grades at the end of the semester compared to their peers who scored higher on the pretest of spatial visualization. Thus, research consistently supported the importance of promoting and training students' spatial skills as a potentially effective way to increase their chances of academic success in their secondary, professional, and university STEM education [25,26].

The relevance of spatial skills in engineering degrees is particularly evident in subjects such as Graphic Expression, which are fundamental mandatory courses in the first year of many engineering programs. The main objective of these courses is to familiarize students with graphic language as a mean to facilitate the conception and study of shapes, design, and development of projects, as well as the exchange of information among engineering professionals. The terms Graphic Expression and 'Technical Drawing' are often used interchangeably in various contexts, although they do not refer exactly to the same thing [27]; Graphical Expression encompasses both artistic and Technical Drawing, although the artis-

tic component in engineering tends to have less weight compared to other disciplines, such as architecture or industrial design, because both involve the development and application of visual–spatial and communication skills in various tasks, following standard drawing conventions and rules.

Numerous authors observed that students in technical careers and vocational training programs require spatial skills to succeed academically in Technical Drawing subjects [20,28–32]. Saorín-Pérez et al. [18], in addition to confirming this relationship specifically in engineering degrees, found that studying Technical Drawing improves mental rotation skills [33], while other studies also highlight its positive effects on learning in other engineering subjects [24,34]. In summary, Technical Drawing is closely linked to spatial skills and holds significant relevance in both engineering studies and professional fields. It is evident that there is a close relationship between students' spatial skills and their academic performance in graphic expression subjects, emphasizing its importance in both engineering studies and professional domains.

However, even though knowledge, skills, and attitudes related to Technical Drawing are particularly relevant in engineering studies [33], the structure of the Spanish educational system does not require students in the Scientific–Technical baccalaureate to take Technical Drawing I and Technical Drawing II courses in order to access these university programs. Specifically, the Spanish educational system stipulates that students in the science and technology track, in addition to mandatory mathematics courses, must choose only two subjects in their first year of baccalaureate between Biology and Geology, Physics and Chemistry, Technology and Engineering I, and Technical Drawing I [35]. Similarly, in the second year of this baccalaureate track, students must select only two subjects from Biology, Geology and Environmental Science, Physics, Chemistry, Technology and Engineering II, and Technical Drawing II [35]. As a result, a notable proportion of students entering engineering schools never chose to study Technical Drawing during their years of baccalaureate. Instead, they focus on subjects such as physics and chemistry, which enable access to a wide range of STEM degrees, thereby developing their spatial skills to a lesser extent than desirable for engineering degrees. More specifically, a significant percentage of first-year students in all engineering degree programs at UPCT never took Technical Drawing in their baccalaureate studies (between 8% and 26%, depending on the cohort and specific degree), or only took it during the first year, which greatly hampers their ability to effectively learn and progress in the Graphical Expression subject.

### 1.2. Promoting and Training Spatial Abilities

Previous research also provides ample evidence that spatial skills are malleable and trainable [25,36,37], and significant efforts were made to improve them in order to promote better learning and performance of students in STEM subjects at various levels of the education system. Actions taken prior to university, such as STEM advanced subject matter, STEM college courses while in high school, and STEM competitions, are significantly related to future STEM achievement outcomes, such as STEM occupations and STEM PhDs [21]. It was also noted that numerous universities implement actions aimed at enhancing these skills once students enter university studies, such as offering leveling programs or "0" courses that involve the use of mental skills that were shown to reduce early dropout rates [24].

Furthermore, apart from the experiences and learning support activities developed in formal contexts, informal learning experiences (e.g., play experiences, hobbies, video games, etc.) can also promote the development of spatial abilities [37–41]. Several classic studies demonstrate that both formal and informal learning experiences have a clear effect on the results of psychometric tests that assess spatial skills. This is evident when comparing the results of children, adolescents, and young adults who engaged in a wide range of informal activities, such as playing with puzzles and video games, hobbies such as carpentry or DIY projects [37], practical exercises involving spatial skills [42–45], or those who took explicit training courses in spatial skills [46,47].

In line with this, two meta-analytic studies are of particular interest for this work [25,36]. Uttal et al. [25], based on a comprehensive review of previous research (206 primary studies), analyze the effects of different intervention methods on spatial abilities. The study considers three types of training: (a) indirect training programs through video games, (b) training through semester-long courses involving spatial skills, and (c) specific training in these skills through direct practice with spatial tasks, explicit teaching, or computerized lessons. The results show that spatial skills are moderately malleable (with a medium effect size of 0.47), the effects of interventions persist over time (at least in delayed measurements up to one month), and they transfer to other spatial tasks that were not specifically trained (with a medium effect size of 0.48). Additionally, all three types of training show homogeneous positive effects among themselves. Students with lower spatial skills show greater improvements, and both males and females show similar improvements, although the pretest differences in favor of males are maintained in the posttests. The training is equally effective in childhood, adolescence, and young adulthood. These results are generally consistent with the classic meta-analysis by Baenninger and Newcombe [36], conducted prior to the availability and interest in research on the effects of video game training to improve spatial skills. The previous meta-analysis found that short-term interventions (e.g., single training sessions or multiple brief training sessions lasting less than three weeks) did not produce significantly superior effects compared to mere practice, while interventions with a longer duration (e.g., more than one training session over more than three weeks or training integrated into a curriculum that lasts less than a semester) generated significantly superior effects compared to mere practice and short-term training.

Furthermore, as highlighted by the meta-analysis conducted by Uttal et al., there is a significant body of evidence emphasizing that practice with video games improves spatial rotation abilities [25,37,48,49], which can be beneficial for students in various fields that require this skill [50]. For example, Okagaki and Frensch [51] found that video games such as'Tetris' can enhance performance in mental rotation tasks, while video games such as 'Block Out', which involve mental rotation of 3D pieces, also improve performance in such tasks [52]. Similarly, Subrahmanyam and Greenfield [53] observed that experienced video game players achieved better results in mental rotation tests compared to non-players, and Terlecki et al. demonstrated that continuous training with a specific video game (Tetris) improved spatial rotation ability in university students [50]. However, there are still discrepancies among researchers regarding which characteristics of video games might explain the improvements in spatial skills to a greater extent. Some authors suggest that the observed improvements depend on the correspondence between the skills involved in the video game and the skills evaluated after practice and training with them [51], while others argue that the improvements may be partly due to players' enhanced selective visual attention abilities [48,49]. Nonetheless, research findings highlight that mental rotation can be more efficiently improved through certain video games compared to others [54].

### 1.3. The Present Study

The main objective of this study addresses the issue that was evident in recent years regarding the significant decline in academic performance in the subject of Graphic Expression among first-year engineering students at UPCT. Specifically, the aim is to analyze and compare the effects of a set of variables that previous research showed to be related to academic outcomes in the first year of university and persistence in engineering studies during the sophomore year. Specifically, the study examines the relationship and effects on academic performance in Graphic Expression of various sociodemographic variables (gender and age of university entry) and previous educational factors (university entrance grade, prior educational trajectory) of the students, as well as their video game experience time and spatial abilities upon entering university. Additionally, the study also analyzes their combined explanatory power and the potential specific contribution of each variable to performance in the aforementioned subject.

Based on previous research findings, the following hypotheses were formulated:

(a) A significant relationship will be observed between the variables considered in the study and academic results in the subject of Graphic Expression, which is taught in the first year of engineering studies. More specifically, a significant positive relationship will be observed between students' academic results in the subject and (a.1) their prior academic preparation (determined by their university entrance grade), (a.2) their educational trajectory (determined by whether they took Technical Drawing courses in previous educational stages), (a.3) their habitual experience in using video games that require spatial skills (measured by the number of hours dedicated to them per week), and (a.4) their spatial skills (evaluated using the MRT-A). Additionally, students' sociodemographic variables (gender and age of university entry) will also be significantly related to academic performance. According to previous research, (a.5) males and females will show homogeneous academic results, although males will demonstrate higher scores than females both in their spatial skills and the time dedicated to video games, and (a.6) the age of university entry will be inversely related to academic performance in the subject.

(b) The predictive capacity of the variables considered in the study on academic performance will be significant once the potential effects of students' sociodemographic variables are controlled. The variables of prior preparation and whether or not they took Technical Drawing courses in their educational trajectory will show superior explanatory power and will be included in the resulting final predictive model.

The results will be discussed from the perspective of previous research, as well as the possible measures to be taken (both in previous educational stages and once students are enrolled in university) to promote satisfactory results in Graphic Expression subjects and address potential academic difficulties.

## 2. Materials and Methods

### 2.1. Participants

In the study, 161 newly enrolled students in the academic year 2022–2023 from various engineering degrees at the Technical University of Cartagena (UPCT) participated (GIDI, $n = 24$; GIM = 118; and PCEO = 20). The average age of the participants was 18.01 years (SD = 0.92, range 17–23), with 81.5% of them being male. All of them were full-time university students, and the vast majority of them entered the university through the scientific–baccalaureate/EBAU (96.3%); only 6 students entered through higher degree vocational training programs. Additionally, 72% of them took Technical Drawing subjects in previous educational stages.

### 2.2. Measures

*Spatial skills.* Spatial skills were evaluated using the Peters et al. Mental Rotation Test (MRT-A) [39], an adaptation of the original paper and pencil Mental Rotation Test by Vandenberg and Kuse [55], in which figures were redrawn using computer-aided design software. Each item consists of a model figure along with four additional figures that need to be mentally rotated to determine if they match or not with the first one, with only two of them corresponding to the same (Figure 1). There were 24 questions divided into two blocks of 12 questions each. Participants were given 3 min to complete each block, with a 2 min rest period in between. An answer was scored as correct only if the two figures matching the model figure were identified. Scores ranged from 0 to 24, representing the total number of correct answers. This test was widely used in previous research, including in Spain, to assess spatial ability and the results of interventions aimed at improving it [18,56]. Both previous research and this study provide evidence of adequate psychometric characteristics (internal consistency = 0.80; test-retest reliability in this study = 0.83).

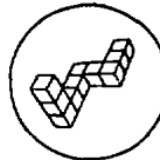 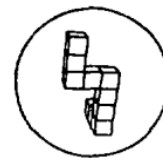 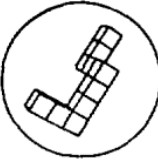 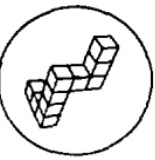 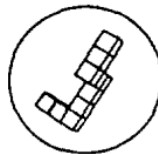

**Figure 1.** MRT-A test sample item [39].

*Video game time.* The total number of weekly hours that participants spend on different types of video games involving spatial skills was recorded in this study (e.g., first-person shooter or FPS games such as Fortnite, 3D construction games such as Minecraft, Tetris, Tetris 3D, and first-person role-playing games or FPRPGs such as Elden Ring).

Additionally, the study also considered various sociodemographic variables (gender and age) and educational variables (mode of university admission grade, previous educational trajectory, and study dedication) of the participants, which were obtained through a questionnaire specifically designed for this purpose. At the end of the academic year, the participants' grades in Graphic Expression, a common mandatory subject in the first year of all engineering degrees at UPCT, were collected as the response variable.

Regarding the explanatory variables, categorical variables were encoded as follows: gender (0 = male, 1 = female); educational trajectory, determined by whether or not *Technical Drawing* was taken in previous educational stages (0 = no, 1 = yes); study dedication, defined as the study regimen (0 = part-time, 1 = full-time), and mode of admission or pathway through which they entered the university (0 = Vocational Training; 1 = Scientific-Technological Baccalaureate). As for the continuous explanatory variables, the participants' *Age* of *Admission* is defined by the years they completed upon entering the university and commencing this study, and the *University Admission Grade* is defined as the numerical score obtained after taking the university entrance exams, expressed on the Spanish educational system scale ranging from 0 to 14 points.

Regarding the response or criterion variables, firstly, the variable *Academic Performance* in Graphic Expression is considered, defined by the final score obtained by students on a scale of 0–10. Secondly, based on the previous variable, the categorical variable pass/fail in the subject is defined (where 0 = pass, 1 = fail). Basic descriptive statistics and the bivariate correlation matrix between the explanatory variables and the criterion variables of the study are shown in Table 1, while the proportions of the explanatory categorical variable modes on the pass/fail variable, as well as the mean and standard deviation of the quantitative variables (*Age* and *University Admission Grade*), can be found in Table 2.

### 2.3. Procedure

First, approval for the development of the study was obtained from the Ethics Committee of Technical University of Cartagena (protocol code CEI22_005 2022/06/13). Participation in the study was proposed for all students enrolled in the freshman bachelor's degree in mechanical engineering (GIM), bachelor's degree in design engineering (GIDI), and double degree in mechanical engineering and design (PCEO) at UPCT during the 2022–2023 academic year, achieving a potential sample of 246 students. As previously mentioned, 161 students (65.45%) finally participated in the survey. Participation was voluntary, and in accordance with the Declaration of Helsinki, participants provided informed consent after being presented with the study objectives.

The MRT-A was administered the first week of September 2022, prior to the start of the Graphic Expression classes. After completing the MRT-A, participants also completed a questionnaire with sociodemographic and educational variables and time spent playing video games involving spatial skills. The academic performance of the students was obtained at the end of the academic year.

*2.4. Statistical Analysis*

Firstly, since all participants reported full-time study dedication, and almost all of them (except for six students) entered university through the baccalaureate–EBAU pathway, both variables were excluded from the statistical analyses. Considering this aspect, a bivariate descriptive analysis was first conducted to explore the relationship between the explanatory and response variables considered in the study. Subsequently, multiple regression analyses were performed to analyze the predictive capacity of the explanatory variables on the two criterion variables. All statistical analyses were conducted using SPSS v. 26.0 (SPSS Inc., Chicago, IL, USA).

The predictive capacity of the explanatory variables on academic performance in Graphic Expression was analyzed through multiple linear regression. In this case, categorical explanatory variables were introduced into the predictive model using dummy coding. On the other hand, given the dichotomous categorical nature of the criterion variable (pass/fail in the subject), a logistic regression model was applied to determine the odds ratio and relative risk of not passing the subject compared to passing it based on the levels of the categorical explanatory variables. According to this model, the logit was, in our case, the natural logarithm of the odds of belonging to category 1 (failing the subject) versus category 0 (passing the subject) of the response variable.

**3. Results**

*3.1. Preliminary Analyses*

Table 1 presents basic descriptive statistics and the bivariate correlation matrix between the explanatory variables and the criterion variables of the study.

**Table 1.** Descriptive statistics and correlations for study variables.

| Variables | M | SD | 1 | 2 | 3 | 4 | 5 | 6 | 7 | 8 |
|---|---|---|---|---|---|---|---|---|---|---|
| 1. Gender [a] | 0.19 | 0.38 | - | | | | | | | |
| 2. Age | 18.01 | 0.93 | −0.04 | - | | | | | | |
| 3. Technical Drawing taken [b] | 0.72 | 0.45 | −0.02 | −0.21 ** | - | | | | | |
| 4. University Admission Grade | 10.01 | 1.83 | 0.09 | −0.24 ** | 0.34 *** | - | | | | |
| 5. Video game time | 4.79 | 4.34 | −0.26 *** | −0.01 | −0.02 | 0.17 * | - | | | |
| 6. Spatial skills (MRT-A) | 12.17 | 5.30 | −0.05 | −0.05 | 0.22 ** | 0.17 * | 0.04 | - | | |
| 7. Academic performance | 4.92 | 2.81 | 0.12 | −0.26 *** | 0.37 *** | 0.46 *** | −0.07 | 0.21 ** | - | |
| 8. Pass/fail [c] | 0.37 | 0.48 | −0.13 | 0.19 * | −0.28 *** | −0.41 *** | 0.07 | −0.17 * | −0.81 *** | - |

[a] 0 = male and 1 = female. [b] 0 = no and 1 = yes. [c] 0 = pass and 1 = fail. * $p < 0.05$, ** $p < 0.01$, *** $p < 0.001$.

University *Admission Grade*, having taken *Technical Drawing*, and *Spatial Skills* show significant positive correlations with the academic performance variables. Video game time is significantly positively related to *University Admission Grade*, but not to the criterion variables. Regarding sociodemographic variables, participants' age shows a significant inverse relationship with various educational variables prior to university admission (University Admission Grade and Technical Drawing taken), as well as with the criterion variables. Gender variable shows a significant inverse relationship with the video game time variable ($r = -0.26$, $p < 0.001$), with women spending significantly less time on this activity (males, M = 4.51, DS = 4.40; females, M = 1.38; DS = 3.2; mean difference = −3.12, $t_{92,772} = -5.08$, $p < 0.001$, $d = -0.74$ IC 95% [−1.14,−034]). Additionally, males and females have similar scores on the MRT-A, whether or not they took Technical Drawing in previous educational stages or whether they do or do not spend time on videogames, with males showing slightly higher scores than females (males, M = 12.29, DS = 5.32; females, M = 11.67; DS = 5.26; and mean difference = −0.62, $t_{160} = 0.58$, $p > 0.05$, and $d = -0.11$, IC 95% [−0.51, 0.28]). On the other hand, participants who took Technical Drawing in previous educational stages show significantly higher academic achievement in Graphic Expression than their peers who did not ($M_{NO} = 3.16$, $DS_{NO} = 3.06$; $M_{YES} = 5.56$; $DS_{YES} = 2.51$; mean difference = −2.40, $t_{160} = -5.07$, $p < 0.001$, $d = 0.90$, IC 95% [0.52,1.25]), as is also observed in mental rotation skills ($M_{NO} = 10.31$, $DS_{NO} = 5.52$; $M_{YES} = 12.84$; $DS_{YES} = 5.05$: mean

difference = $-2.53$, $t_{160}$= $-2.78$, $p < 0.01$, $d = 0.49$ IC 95% [0.14,0.83]). The set of significant correlations shows values ranging from low to moderate low, with the highest magnitude correlation being between University Admission Grade and Academic Performance in Graphic Expression (r = 0.46, $p < 0.001$).

### 3.2. Predictive Capacity on Academic Performance in Graphic Expression

Table 2 shows the results of the multiple regression analysis on academic performance in the subject of 'Graphic Expression'. The proposed model significantly predicts the criterion ($F_{6156} = 11.39$, $p < 0.001$), demonstrating a multiple correlation of 0.56 with the criterion and explaining 31.3% of its variance. In the resulting regression equation, with predictors listed in descending order of predictive capacity, the variables *University Admission Grade* (β = 0.37, $p < 0.001$) and having taken *Technical Drawing* in previous educational stages (β = 0.20; $p < 0.01$) are included, while the *Age* of the participants shows a value close to statistical significance (β = $-0.13$, $p = 0.065$).

**Table 2.** Multiple regression analysis on academic performance in the subject.

| Variable | B | SE B | t | β | p | 95% CI |
|---|---|---|---|---|---|---|
| Constant | 5.07 | 4.33 | 1.17 | | 0.243 | [−3.48, 13.63] |
| Gender | 0.51 | 0.52 | 0.97 | 0.07 | 0.335 | [−0.52, 1.54] |
| Age | −0.41 | 0.22 | −1.86 | −0.13 | 0.065 | [−0.84, 0.03] |
| Technical Drawing taken | 1.28 | 0.48 | 2.68 | 0.20 | 0.008 | [−0.34, 2.23] |
| University Admission Grade | 0.57 | 0.12 | 4.82 | 0.37 | 0.001 | [0.34, 0.81] |
| Video game time | −0.05 | 0.03 | −1.53 | −0.11 | 0.129 | [−0.11, 0.02] |
| Spatial skills (MRT-A) | 0.51 | 0.04 | 1.34 | 0.18 | 0.181 | [−0.02, 0.13] |

### 3.3. Predictive Capacity on Pass/Failing the Subject

Table 3 displays the proportions of failures conditioned by each of the modalities of the categorical explanatory variables, as well as the mean (*M*) and standard deviation (*SD*) of the quantitative variables in both groups of students. The total number of participants who failed the subject was 59, representing a total of 36.42% of them. Preliminary analyses reveal a significant inverse relationship between failing the subject and having taken *Technical Drawing* ($\chi^2$ (1) = 12.86; $p < 0.001$), with the odds of failing versus passing the subject being 3.63 times higher for students who did not take Technical Drawing compared to those who did—95% CI [1.76–7.49]. However, there was no significant relationship with gender ($\chi^2$ (1) = 2.82; $p > 0.05$), where the odds of failing were 2.16 times higher for males than for females—95% CI [0.87, 5.40]. As for the continuous explanatory variables, students who failed the subject were significantly older than their peers who passed (mean difference = 0.37, $t_{72.83}$ = 2.1, $p < 0.05$, $d = -0.40$, IC 95% [−0.73, −0.08]), had lower university admission grades (mean difference = $-1.54$, $t_{159}$= $-5.58$, $p < 0.001$, $d = 0.92$, IC 95% [0.59, 1.26]), and lower spatial abilities (mean difference = $-1.84$, $t_{159} = -2.14$, $p < 0.05$, $d = 0.36$, IC 95% [−0.02, −0.37]). There were no significant differences between students who failed and those who passed the subject in terms of weekly hours dedicated to video games (mean difference = 0.90, $t_{159} = 0.87$, $p = 0.19$, $d = -0.14$, IC 95% [−0.46, 0.18]).

The resulting predictive model for the criterion of failing the subject shows a significant overall fit ($\chi^2$ (6)= 38.1, $p < 0.001$), explaining 29.4% of the criterion's variance (Nagelkerke's R2). The model correctly classifies 68.8% of cases in the response variable (Hosmer and Lemeshow $\chi^2$ (8) = 5.48; $p < 0.71$), showing a sensitivity of 62.1% in detecting subjects who ultimately fail the subject, and a specificity of 74.7% in correctly identifying subjects who ultimately pass it. As shown in Table 4, the process of statistical modeling resulted in a final model that includes only significant main effects of the variable University Admission Grade, highlighting that for every 1-point increase in the University Admission Grade scale, the odds of failing the subject multiply by 0.61.

**Table 3.** Descriptive statistics and relationship with passing/failing the subject.

| Variables | $p_i$ | $p_i$ (Fail | $x_i$) |
|---|---|---|
| Gender | | |
| Men | 52 | 39.7 |
| Women | 7 | 23.3 |
| Technical Drawing taken | | |
| Yes | 33 | 28.2 |
| No | 26 | 59.1 |
| Age | M = 18.01 S = 0.9 | $M_{FAIL}$ = 18.3 $S_{FAIL}$ = 1.3 $M_{PASS}$ = 17.9 $S_{PASS}$ = 0.6 |
| University Admission Grade | M = 10.01 S = 1.8 | $M_{FAIL}$ = 9.0 $S_{FAIL}$ = 1.7 $M_{PASS}$ = 10.6 $S_{PASS}$ = 1.6 |
| Video game time | M = 4.79 S = 4.3 | $M_{FAIL}$ = 5.4 $S_{FAIL}$ = 5.8 $M_{PASS}$ = 4.5 $S_{PASS}$ = 3.4 |
| Spatial skills (MRT-A) | M = 12.2 S = 5.3 | $M_{FAIL}$ = 11.0 $S_{FAIL}$ = 5.3 $M_{PASS}$ = 12.9 $S_{PASS}$ = 5.2 |

**Table 4.** Results of the logistic regression on the variable pass/fail.

| | Exp ($\beta$) | IC 95% Exp ($\beta$) | $x^2$ Wald | gl | *p* |
|---|---|---|---|---|---|
| Intercept | 0.90 | | 0.01 | 1 | 0.98 |
| Gender * | | | | | |
| Male | 1.96 | [0.67–5.72] | 1.52 | 1 | 0.22 |
| Technical Drawing taken * | | | | | |
| No | 1.83 | [0.78–4.30] | 1.94 | 1 | 0.16 |
| Age | 1.24 | [0.80–1.92] | 0.90 | 1 | 0.34 |
| University Admission Grade | 0.61 | [0.48–0.78] | 15.71 | 1 | 0.001 |
| Spatial skills—MRT-A | 0.97 | [0.90–1.03] | 1.01 | 1 | 0.32 |
| Video game time | 1.04 | [0.98–1.10] | 1.63 | 1 | 0.20 |

* Reference categories: Gender→Female; Technical Drawing taken→Yes.

## 4. Discussion

In this study, the relevance of various socio-personal and educational variables of engineering students was analyzed to explain the poor academic outcomes observed in the subject of Graphic Expression, a mandatory course in the first year of various engineering degrees at UPCT (GIM, GIDI, and CPEO). This study is built upon extensive previous research (a) on the determinants of academic success in the first year of university, particularly in STEM and engineering disciplines, (b) on the relevance and effects of spatial skills on academic performance in Technical Drawing and graphic expression subjects in secondary and tertiary education, and (c) on learning experiences and educational interventions, both in formal and informal educational contexts, that facilitate the development of spatial skills.

Firstly, bivariate relationships and the effects on academic outcomes in the subject of Graphic Expression of a set of variables that previous research showed to be related to academic success in the first year of engineering studies and sophomore year retention were analyzed. Specifically, socio-demographic variables (gender and age at university enrollment) and previous educational factors (university entrance grade and educational trajectory) of the students were considered, as well as their video game experience time and spatial skills. Secondly, through linear regression and multiple logistic regression techniques, the joint explanatory capacity of these variables, as well as the relative and specific contribution of each of them on academic outcomes in Graphic Expression were determined. These outcomes were assessed based on both the final grade in the subject and whether or not the student passed the course. Subsequently, following the working hypotheses, the main results and conclusions of the study are highlighted, emphasizing their relationships and contributions to previous research, as well as potential educational measures to be considered based on these findings in order to promote academic success in Graphic Expression in engineering studies.

(a) A significant relationship between the variables considered in the study and the academic outcomes in Graphic Expression will be evidenced. Thus, a significant positive relationship will be observed between students' academic outcomes in this subject and (a.1) their prior preparation (determined by their university entrance grade), (a.2) their educational trajectory (determined by whether or not they took Technical Drawing subjects in previous educational stages), (a.3) the time dedicated to video games that involve the use of spatial skills (number of weekly hours spent on them), and (a.4) their spatial skills (evaluated through the MRT-A). Regarding the students' sociodemographic variables, (a.5) males and females will show homogeneous academic results, although males will demonstrate higher scores than females both in their spatial skills and the time dedicated to video games, and (a.6) the age of university enrollment will be inversely related to academic outcomes in the mentioned subject.

The results partially confirm this initial hypothesis. Thus, (a.1) prior academic preparation shows a significant positive relationship with academic performance in Graphic Expression and a negative relationship with failing the course; (a.2) similar results are evident when considering participants' previous educational trajectory; (a.3) however, the time spent on video games involving the application of spatial skills does not show a significant relationship with the outcomes in the subjects; (a.4) spatial skills also demonstrate a significant positive relationship with performance in graphic expression and a negative relationship with failing the course.

These results coincide mainly with those highlighted in previous research, where students' previous preparation [2,4–14], educational trajectory [11,24,33], cognitive abilities, and specifically in this study, spatial rotation abilities [17–19,22–26], are significantly positively related to academic outcomes in STEM studies in general, and engineering in particular. However, it is important to note that these significant correlations show values ranging from low to moderate, with the most specific variable (mental rotation abilities) showing a weaker correlation with the criterion, while the broader variable (prior preparation) demonstrates a stronger relationship, followed by students' educational trajectory (whether or not they took technical drawing subjects). It is noteworthy that the variable of time spent on video games, contrary to what was highlighted in the working hypotheses, does not show any significant relationship with performance in graphic expression. Recent systematic reviews [27] and meta-analytical studies [25,36] demonstrated that both informal learning experiences (e.g., playing with puzzles and video games, hobbies, and crafts) and formal learning experiences (e.g., explicit teaching on spatial tasks and strategies and indirect training programs using video games) promote the development of spatial abilities [37–41]. Therefore, it was hypothesized that the extent of experience with video games would be related to academic outcomes in graphic expression. However, although the results lead to the rejection of this hypothesis, they also demonstrate a significant positive relationship between video game experience, prior academic preparation (university entrance grade), and spatial rotation abilities (MRT-A) of the students. This coincides with the results of previous research and suggests the possibility that the effects on academic outcomes may not be direct, but rather, indirect, and mediated by these latter two variables.

Regarding the effects of students' sociodemographic variables, the results also partially confirm the working hypotheses. Firstly, (a.5) no significant relationships are observed between participants' gender and their performance in graphic expression or their scores in the MRT-A, while a significant relationship is evident with the time dedicated to video games, with males spending more time on them.

More specifically, similar to what previous research with first-year engineering students highlights [2,7,10,57–59], no significant differences are observed between males and females in their performance in graphic expression, and there is even a tendency for females to achieve higher results than males. On the other hand, contrary to what various review studies and the working hypotheses [37] suggest, no significant relationship is found between participants' gender and their spatial abilities. However, a considerable volume of studies show that males exhibit higher spatial abilities compared to females [60–62],

which some authors explain in terms of biological factors (e.g., prenatal hormone gender differences, as well as functional and structural gender differences in brain areas involved in mental rotation) [63,64], while others attribute it to environmental factors (e.g., play experiences and gender stereotypes) [65–67]. Additionally, some authors emphasize that these differences should be interpreted with caution, especially regarding their effects on performance in STEM fields [39], and note that in recent years, the differences between men and women decreased [18]. In any case, since all researchers agree that spatial abilities can be improved through informal learning experiences and explicit training, the homogeneity between males and females in their prior academic competencies (university entrance grade), the educational path pursued (scientific–technological track in high school), and the similar proportions of both genders that took technical drawing subjects (72.5% of males and 70% of females) would justify them exhibiting homogeneous levels of spatial abilities upon entering engineering and in their academic performance in graphic expression. On the other hand, these study results highlight that males have significantly higher video game experience than females, which aligns with previous research findings [50] suggesting that the differences favoring males in spatial abilities may be partially mediated by their greater experience with video games.

Lastly, in line with these initial hypotheses and previous research [11], (a.6) the age of entering engineering studies shows a significant inverse relationship with academic performance in graphic expression. This is expected, since late entry into studies is often associated with various reasons, such as problematic academic trajectories (e.g., grade repetition, failed subjects, and vocational indecision) and/or a lack of alignment with the demands of the studies (e.g., not having taken technical drawing subjects).

(b) The explanatory and predictive capacity of the variables considered in the study on academic performance will be significant once the potential effects of students' sociodemographic variables are controlled. Prior preparation and whether or not they took technical drawing courses in their educational trajectory will show superior explanatory power and will be included in the resulting final predictive model.

The results also basically confirm this second working hypothesis, indicating that the set of variables considered in the study significantly predicts the outcomes in Graphic Expression, whether the criterion is operationalized in terms of academic performance (final grade in the subject, on a scale of 0–10) or in terms of passing or failing the subject (pass/fail), explaining approximately one-third of its variance in both cases. However, it should be noted that the variables introduced in the resulting regression models differ depending on how the academic outcomes are operationalized (final grade in the subject and passing or failing the subject). Thus, the variables with the greatest predictive capacity and that offer a specific contribution to explaining academic performance, in order of importance, are university entrance grade (previous preparation) and the educational trajectory of the students (whether they took technical drawing in previous stages), without sociodemographic variables, scores on mental rotation skills, and video game dedication time being introduced into the regression equation. On the other hand, in the predictive model for passing or failing the subject, only the variable of previous academic preparation is introduced, without the remaining variables considered, providing an additional specific contribution to explaining this criterion.

First, it should be noted that the bivariate relationships between the explanatory variables and both criteria already pointed to these results, as previous preparation and educational trajectory showed the strongest correlations with both academic performance and passing or failing the subject. Age of university entrance (inversely) and mental rotation skills also show significant relationships with both performance criteria, although the intensity of the relationship is lower. Similarly, the preliminary analyses for logistic regression also provide clarification: students who did not take technical drawing in previous educational stages had an odds ratio of failing graphic expression 3.63 times higher compared to their peers who did take it; additionally, the effects of university entrance grade on passing or failing the subject were large ($d = 0.92$), while the effects of

age of university entrance and spatial abilities assessed through the MRT-A were small (in both cases, d < 0.41).

In summary, the high magnitude of the effect of previous academic preparation on the outcomes in graphic expression coincides with the results and conclusions of numerous investigations carried out in a wide variety of university studies [68,69], and specifically in engineering [2,15,16]. These results can be argued from various perspectives: (a) greater previous academic preparation (skills and prior knowledge) facilitates meaningful learning of new knowledge; (b) students' previous academic results are a result of their involvement, dedication, and effort in studying over time; (c) achieving satisfactory academic results throughout the education system promotes maintaining, if not increasing, high levels of commitment to learning; and (d) previous academic achievements are related to subsequent academic achievements, as both are affected by students' intellectual abilities, which are relatively stable personal characteristics [70]. Otherwise, academic preparation constitutes a holistic variable that encompasses the effects of students' cognitive, personal, motivational, and experiential dimensions throughout the education system, and consequently, its contribution and effects on academic outcomes are superior to those evidenced by more specific skills. On the other hand, the educational trajectory, defined in this study as whether students took technical drawing subjects in previous educational stages or not, is also incorporated into the resulting predictive model for academic performance in graphic expression, but not for passing or failing the subject. The significant and specific positive effects of previous training in Technical Drawing on the outcomes in Graphic Expression also coincide with research conclusions, both due to the close correspondence between the two subjects [27] and the evidence that spatial skills can be improved through semester-long courses involving spatial skills [25,36], improvements that persist over time and even transfer to other spatial tasks not directly trained.

However, it may come as a surprise that whether students took Technical Drawing or not in previous educational stages is not incorporated into the resulting predictive model for passing or failing the Graphic Expression subject, despite the intense bivariate relationship between them. These results can be explained by considering at least two complementary aspects:

(a) The engineering students participating in the study have similar educational trajectories (scientific–technological high school, all of them taking mathematics and STEM elective subjects), although they may differ in whether they took Technical Drawing among the optional subjects. In other words, they all possess general reasoning abilities, as well as mathematical and verbal skills that allowed them to meet the national STEM content standards to enroll in these university studies. On the other hand, while several studies found that spatial abilities offer a significant specific contribution to STEM students' academic outcomes after controlling for the effects of their mathematical and verbal abilities [21–23], it is also true that spatial abilities were shown to be much less critical than these latter abilities, even in mechanical engineering subjects [21–23]. Ultimately, despite not taking Technical Drawing in previous stages, students may overcome their initial deficiencies in spatial abilities and technical drawing knowledge through monitoring, dedication, and involvement in the Graphic Expression subject, ultimately succeeding in it.

(b) On the other hand, we cannot forget that the university entrance year is particularly relevant and complex, given the numerous academic, personal, and social changes that students must face, which directly affect their academic results. Extensive meta-analytic studies [8,9], as well as research specifically conducted in engineering [2,7,10,15,16,59,71–73], confirm this issue, highlighting that university students' experiences, perceptions, and attitudes mediate the effects of personal and pre-university variables on academic success in the first year of engineering studies, as well as on persistence in studies in the sophomore year. More specifically, studies focused on engineering emphasize the relevance of (a) pre-university preparation, but also (b) socialization experiences with engineering in earlier stages of the educational system, (c) interest in engineering and other scientific fields, (d) self-efficacy in mathematics, sciences, and computer science, (e) learning

and study strategies, (f) economic problems, (g) family support, (h) interactions with classmates and professors, and (i) the initial level of institutional and degree commitment [2,7,10,15,16,59,71–73].

In any case, the results of the logistic regression analysis for passing or failing the subject do not diminish the relevance of taking Technical Drawing in previous stages of the education system, which was previously evidenced by its significant and specific effects on academic outcomes in Graphic Expression, even after considering the effects of previous preparation. On the other hand, the measures to consider for promoting students' academic success once they enter university should focus on those aspects that are susceptible to intervention and improvement, considering and building upon their abilities and prior knowledge before university entrance. This is the case with spatial abilities, which can be intervened upon and improved through various means, with the effects of the intervention persisting in the medium to long term [25,36].

The results obtained in this study have significant implications, both in institutional and instructional terms, to be considered both prior to university admission and once students are already enrolled in engineering studies.

Firstly, a fundamental conclusion is that universities need to increase and strengthen the programs and actions they develop in earlier stages of the education system in order to (1) attract students with high skills and relevant knowledge in engineering fields, (2) increase the knowledge that high school students and their families have about these studies and the breadth of professional fields they encompass, and (3) promote greater motivation and interest among high school students towards engineering degrees [2]. More specifically, this refers to: (a) identifying students with higher abilities, performance levels, and self-efficacy in mathematics, technology, and drawing at early stages of secondary education (and even primary education); (b) increasing educational experiences and doses, as defined by Wai et al. [21], aimed at socializing and enhancing high school students' motivations and knowledge about engineering; (c) if possible, linking these actions to the specific motivations and interests of students related to various engineering disciplines, such as mechanics, electronics, robotics, or programming [74]; and (d) enhancing the relationship and connections between engineering degrees and educational guidance services and STEM teachers in secondary education, especially in the Scientific–Technological Baccalaureate, and particularly with those who teach elective subjects (Technical Drawing, Technology and Engineering, Biology, Geology and Environmental Science, Physics, and Chemistry). Otherwise, enabling high school counseling and guidance services, and teachers in earlier stages, to have a more comprehensive and realistic understanding of engineering degrees more directly related to the subjects they teach (skills and knowledge involved, organization of studies, level of difficulty, professional fields, etc.) offers their students a more complete and realistic view of them. However, we are aware that putting this latter aspect into practice is more complex than it may seem, among other considerations, due to the vast multiplicity of existing engineering degrees (over 100 different degrees in our context), which greatly hinders guidance services, teachers, students, and their families from having precise knowledge of them and making decisions about the educational path that may be most suitable for them.

Secondly, once students entered engineering programs, the results highlight the importance of identifying as early as possible those students with lower levels of prior preparation (e.g., mathematical and spatial skills) and/or who followed an educational path with deficiencies in the most relevant skills and knowledge for specific engineering degrees (e.g., Technical Drawing) in order to address the knowledge gaps they may have. For example, strategies such as implementing Early Warning Systems in universities [75], mentorship programs involving faculty and/or peer-mentoring systems [76], or supplementary developmental education programs, courses, and services [77] can facilitate the detection and promotion of skills in which students show the greatest deficiencies. Given their close relationship with academic outcomes in the first year of university, it is also essential to consider other intervention strategies [2], such as the development of training courses

and workshops aimed at improving students' learning and study strategies, including seeking help from teachers and peers, avoiding procrastination, reducing test anxiety, and increasing effort regulation in the form of persistence plans when facing difficult tasks [78].

Thirdly, it is important to highlight the role of teachers' behaviors in the first year at university, emphasizing the relevance of considering teaching methodologies and designing classroom environments that help students develop effective learning and problem-solving strategies and support student engagement. This includes aspects such as classroom goal structures (mastery-oriented classrooms), learning tasks (e.g., authentic, challenging, open tasks, meaningfulness, and relevance of activities), teachers' instructional strategies and practices (e.g., using collaborative learning techniques, instructional and motivational scaffolding, and effective feedback), and interpersonal relationships in the classroom (e.g., teacher and student/peer support, mutual respect, support for autonomy, competence, and collaboration) [21,79].

We are aware that this study has several limitations, including the need to consider larger samples of students from different universities and engineering programs, the inclusion of variables that previous research also identified as relevant for predicting success in university studies (e.g., family socioeconomic level, vocational indecision, and degree preferences), expanding the range of spatial skills considered, and designing longitudinal research designs that allow for examining the effects of Technical Drawing and Graphic Expression courses and workshops on these skills. These issues will be addressed in future work, expanding the range of personal and institutional variables considered and using structural analysis techniques to determine their direct and indirect effects on students' learning outcomes. Nevertheless, both conceptually and practically, the results of the study reaffirm and expand upon the conclusions of previous research, highlighting the relevance and predictive capacity of the variables considered (sociodemographic, academic preparation, educational path, mental rotation skills, and video game usage on the outcomes of Graphic Expression in engineering studies. They provide further clarification and specificity for first-year engineering students in the Spanish context. From an applied perspective, they emphasize the need for universities to establish action plans in both earlier stages of the education system and the first year of university, aimed at preventing and intervening in potential difficulties students may face during their transition to university, particularly in the area of Graphic Expression. This would allow for the quick identification of students in need of specific educational support in this area and the enhancement of students' spatial skills beyond the development of the course material.

## 5. Conclusions

This study provides evidence of the special relevance and predictive capacity of prior academic preparation and the educational trajectory (whether or not technical drawing was taken) on academic outcomes in Graphic Expression, a mandatory subject in the first year of engineering studies. It also found a positive relationship between students' mental rotation skills and learning outcomes in the subject. The time spent playing video games is not related to students' academic performance in the subject, although it does positively correlate with their spatial abilities. There were no significant differences in academic results or spatial abilities based on gender, supporting the idea that formal and informal learning experiences (such as educational trajectories and university admission scores) reduce potential gender differences in spatial abilities, as highlighted in various meta-analytical studies. This reinforces the importance of promoting the development of spatial skills throughout the education system.

Despite the evidence of the relationship between students' spatial abilities and their academic achievements in STEM fields, particularly in engineering, further studies are needed to analyze the extent to which interventions and improvements in spatial abilities translate into improved academic outcomes for students [80]. Studies conducted in introductory engineering courses clearly point in this direction [19,24,81–83], although it should not be forgotten that, in any case, educational interventions on spatial skills may generate

moderate significant effects on academic performance and persistence in STEM studies, while also expanding opportunities for a large number of students who might otherwise give up on accessing these degrees or fail to obtain their degrees after accessing them.

**Author Contributions:** Conceptualization, J.S.V., F.C., M.C.F. and R.G.-R.; methodology, J.S.V. and R.G.-R.; software, F.C.; validation, J.S.V., F.C., M.C.F. and R.G.-R.; formal analysis, M.C.F. and R.G.-R.; investigation, F.C. and M.C.F.; resources, J.S.V. and R.G.-R.; data curation, F.C.; writing—original draft preparation, J.S.V., F.C., M.C.F. and R.G.-R.; writing—review and editing, J.S.V. and R.G.-R.; visualization, M.C.F.; supervision, J.S.V.; project administration, R.G.-R.; funding acquisition, J.S.V. and F.C. All authors have read and agreed to the published version of the manuscript.

**Funding:** This research received no external funding.

**Institutional Review Board Statement:** The study was conducted in accordance with the Declaration of Helsinki, and approved by the Ethics Committee of Technical University of Cartagena (protocol code CEI22_005 2022/06/13) for studies involving humans.

**Informed Consent Statement:** Informed consent was obtained from all subjects involved in the study.

**Data Availability Statement:** The data presented in this study are available on request from the corresponding author. The data are not publicly available due to privacy reasons.

**Acknowledgments:** The research work reported here would not have been possible without the participation of those GIM, GIDI and PCEO first year students who volunteered to be part of it, even knowing that no reward would be provided. Many thanks to them all. We would also like to thank Michael Peters, from the Department of Psychology of University of Guelph (Canada), for so kindly providing us the copy of the MRT-A test.

**Conflicts of Interest:** The authors declare no conflict of interest.

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
