# Peer review of "Educational Pathways, Spatial Skills, and Academic Achievement in Graphic Expression in First Year of Engineering"

_education, doi:10.3390/educsci13070756_

Round 1

Reviewer 1 Report

Edit:

-Table 2 should not be referenced before Table 1

Questions:

-how were students solicited to participate? In-class, email, ?

-Were video game hours self-reported values?

Concerns:

In the interventions discussion on lines 567-588, these are based on literature results as there are no interventions included in this study. This seems speculative as there is no data for this study to support these suggestions. This seems especially true for the actions suggested for prior experiences (lines 620-622).

There are minor word usage issues, it might be good to have someone read through and clean up some issues, for example:

-line 218, the word redraw should be redrawn.

-also it is best practice to identify abbreviations when they are first presented. Line 209 has several that are not defined until much later in the paper: GIDI, GIM, PCEO.

Author Response

Please find the reply in the attached file

Reviewer 2 Report

Generally well structured and an interesting topic, although the conclusions are a bit weak in that the results either reinforce things that the article states are already well evidenced, or the hypotheses are not supported by the results. While this must sometimes be the case with research, it leads to a paper that does not have significant impact on knowledge and understanding in the area, particularly given its length. If these results are to be reported, it would perhaps be better to do so in a much briefer article.

Remove second "on" in sentence, line 37-38.

Author Response

(The authors gave the same response as above.)
